# Titanium Surface Characteristics Induce the Specific Reprogramming of Toll-like Receptor Signaling in Macrophages

**DOI:** 10.3390/ijms23084285

**Published:** 2022-04-13

**Authors:** Zaira González-Sánchez, Victoria Areal-Quecuty, Alvaro Jimenez-Guerra, Daniel Cabanillas-Balsera, Francisco Javier Gil, Eugenio Velasco-Ortega, David Pozo

**Affiliations:** 1Cellular and Molecular Neuroimmunology Laboratory, CABIMER-Andalusian Center for Molecular Biology and Regenerative Medicine, University of Seville-UPO-CSIC, 41092 Seville, Spain; zaira.gonzalez@cabimer.es (Z.G.-S.); victoria.areal@cabimer.es (V.A.-Q.); 2Department of Medical Biochemistry, Molecular Biology and Immunology, School of Medicine, University of Seville, 41009 Seville, Spain; 3Department of Stomatology, School of Dentistry, University of Seville, 41009 Seville, Spain; ajimenez8@us.es (A.J.-G.); dacabanillas@us.es (D.C.-B.); 4School of Dentistry, Universitat Internacional de Catalunya (UIC), 08017 Barcelona, Spain; xavier.gil@uic.es; 5Bioengineering Institute of Technology, Universitat Internacional de Catalunya (UIC), Sant Cugat del Vallés, 08195 Barcelona, Spain

**Keywords:** titanium discs, dental implants, surfaces, immune regulation, macrophage cells, Toll-like receptors

## Abstract

Most of the research on titanium-based dental implants (Ti-discs) is focused on how they are able to stimulate the formation of new tissue and/or cytotoxic studies, with very scarce data on their effects on functional responses by immunocompetent cells. In particular, the link between the rewiring of innate immune responses and surface biomaterials properties is poorly understood. To address this, we characterize the functional response of macrophage cultures to four different dental titanium surfaces (MA: mechanical abrasion; SB + AE: sandblasting plus etching; SB: sandblasting; AE: acid etching). We use different Toll-like receptor (TLR) ligands towards cell surface receptors (bacterial lipopolysaccharide LPS for TLR4; imiquimod for TLR7; synthetic bacterial triacylated lipoprotein for TLR2/TLR1) and endosomal membrane receptor (poly I:C for TLR3) to simulate bacterial (cell wall bacterial components) or viral infections (dsRNA and ssRNA). The extracellular and total LDH levels indicate that exposure to the different Ti-surfaces is not cytotoxic for macrophages under resting or TLR-stimulated conditions, although there is a tendency towards an impairment in macrophage proliferation, viability or adhesion under TLR4, TLR3 and TLR2/1 stimulations in SB discs cultures. The secreted IL-6 and IL-10 levels are not modified upon resting macrophage exposure to the Ti-surfaces studied as well as steady state levels of *iNos* or *ArgI* mRNA. However, macrophage exposure to MA Ti-surface do display an enhanced immune response to TLR4, TLR7 or TLR2/1 compared to other Ti-surfaces in terms of soluble immune mediators secreted and M1/M2 gene expression profiling. This change of characteristics in cellular phenotype might be related to changes in cellular morphology. Remarkably, the gene expression of *Tlr3* is the only TLR that is differentially affected by distinct Ti-surface exposure. These results highlight the relevance of patterned substrates in dental implants to achieve a smart manipulation of the immune responses in the context of personalized medicine, cell-based therapies, preferential lineage commitment of precursor cells or control of tissue architecture in oral biology.

## 1. Introduction

The last few years have witnessed a growing demand for implantable medical devices, mainly for two reasons. On the one hand, important advances in the technological and industrial development of nano- and micro-structured materials were made and, on the other, there is an aging world population with more chronic diseases and greater needs for these alternatives [1,2]. In this sense, the use of bone-anchored titanium devices has become a well-established treatment option for dental implants due to their excellent biocompatibility and osseointegration [3,4,5,6,7]. Osseointegration is considered the final outcome of a physiologically immune response after the embedding process of a titanium implant in order to recede from adjacent tissues [8,9,10]. Usually, follow-up studies reported a low proportion of failure for different titanium-based implants [11]. In recent years, progress has been made in the role that the immune system plays in the context of bone regeneration and how different characteristics of the surface on titanium implants modulate the critical parameters of various immunocompetent cells during bone regeneration [12,13,14,15].

Despite advances in our understanding of the interrelationship between titanium dental implants, bone regeneration, immune mediators and even associated peri-implantitis phenomena, we still have limited information about the molecular mechanisms that operate under physiological immune responses in different Ti-discs. Thus, research efforts have focused on the effects of different titanium implants on macrophage polarization, but under resting conditions, i.e., without immunological stimulation in order to explore the potential of engineered surfaces to promote a balanced osteogenesis [15,16,17,18] and a smart manipulation of M2 [19,20,21] or M1 macrophage responses [22]. However, the effect of different titanium dental implants on the ability to trigger an immune response by key immunocompetent cells has not been addressed to date. This is particularly relevant as growing experimental and epidemiological evidences connect oral biology and systemic diseases [23,24,25,26,27,28]. 

Toll-like receptors (TLRs) are part of the pattern recognition receptors highly expressed in macrophages, which are responsible for identifying pathogen-associated molecular patterns (PAMPs) and damage-associated molecular patterns (DAMPs), the conserved molecules linked with pathogenic microbes and cell/tissue damage, respectively [29]. Each TLR recognizes a precise collection of PAMPs, by which intracellular TLRs located in the membranes of endosomes and lysosomes sense nucleic acids derived from bacteria and viruses; cell surface TLRs mainly recognize microbial membrane components, such as lipids, lipoproteins, and proteins [29,30]. Therefore, PAMP signaling by TLRs is essential for host defense responses to pathogens spreading, although its abnormal activation is also involved in the development of chronic inflammatory and autoimmune diseases [31,32,33].

In this work, we characterize for the first time the functional response to different TLR ligands of macrophage cultures on the surface of four different dental titanium discs (Mechanical abrasion, MA; sandblasting plus etching, SB + AE; sandblasting, SB; and acid etching, AE). For this purpose, we use different TLR ligands towards cell surface receptors—bacterial lipopolysaccharide (LPS) for TLR4; imidazoquiline amine analog to guanosine (Imiquimod) for TLR7; synthetic bacterial triacylated lipoprotein for TLR2/TLR1)—and endosomal membrane receptor (poly I:C for TLR3), in order to simulate bacterial (cell wall bacterial components) or viral infections (dsRNA and ssRNA) [34,35]. Our results are of relevance to understand the link between different titanium-based implants and their impact on macrophage homeostatic innate immune responses. 

## 2. Results and Discussion

### 2.1. Characterization of Ti-Discs

The composition and surface properties of Ti-discs were evaluated by X-ray (XR) microfluorescence, Field Emission Scanning Electron Microscope (FESEM) and confocal microscopy. Figure 1 shows the four different discs and their composition by XR-microfluorescence. The MA, SB + AE and AE treatments did not affect disc composition with a Ti concentration close to 99%. However, the SB surface showed both a reduced titanium concentration (86.23%) and increased aluminum (Al) concentration (13.63%). 

The surface roughness was evaluated by confocal profiling acquiring three different measurements in each disc in order to calculate the Ra values. The discs with MA showed the lowest Ra value (0.0261 ± 0.0123 µm, Table 1) for mean peak-to-valley roughness, representing a smooth surface. SB + AE, SB and AE discs had an average surface roughness of the same order of magnitude (1.2351 ± 0.024; 0.9325 ± 0.522; 0.603 ± 0.1813 µm, respectively) up to 50 times higher compared to discs with MA.

Aluminum (Al) was found in SB samples due to the use of Al_2_O_3_ as an abrasive particle for the sandblasting treatment. After the blasting deformation, some abrasive particles (Al_2_O_3_) may become embedded and contaminate the implant surface. With the use of acid etching, the most superficial layers of the implant surface are removed, decreasing the surface stress and cleaning the surface contaminated by particles leftover from the sandblasting process. At the same time, this process helps in the creation of microcavities on the surface of the implant, generating an added nanometric roughness.

These results were also confirmed by FESEM in order to assess the topography of each type of Ti-based implant studied. As Figure 2 shows, the discs treated with MA showed a smooth and slightly undulated surface with circular parallel scratches, while the surfaces of the other discs were rougher and more irregular. Particularly, the SB discs revealed large peaks and valleys of diverse geometry with numerous flat planes, whereas the surfaces of the other Ti-discs were less spiky. The discs exposed to SB + AE procedures showed the highest roughness and an heterogenous nanoporous surface. The discs subjected to AE presented micropores of different sizes. The surfaces observed in the electron microscopy images matched with those of the 3D representations obtained from optical profilometry in the lower panel of Figure 2.

As is already known, blasting is the technique that allows the achievement of a greater surface roughness, this roughness being smoothed by the subsequent acid etching.

### 2.2. Cell Viability of Resting and TLR-Stimulated Macrophage Cultures in Different Ti-Discs

Although cell viability has been previously studied in discs with different surface treatments [36], there is no information on how it is affected once the immunocompetent cell has to respond physiologically to different TLR stimulations. The viability of RAW 264.7 macrophage cells seeded on different titanium surfaces was evaluated in the presence of different TLR ligands. The stimulation of TLR4, TLR7, TLR3 and TLR2/1 was induced with LPS, imiquimod, poly(I:C) and Pam3Csk4, respectively, and cell viability was assessed by quantifying the lactate dehydrogenase (LDH) release and the total LDH after 24 h. As Figure 3 shows, while there were some differences between discs both in LDH released and in total LDH, these were not statistically significance. Therefore, the exposure of RAW 264.7 cells under physiological stressors to different Ti-based dental implants had no impact in terms of cytotoxicity, highlighting the excellent biocompatibility of the discs. Remarkably, these data indicate that the differences observed in the following experiments related to the immune characterization were not due to an alteration in the proliferation and/or viability of RAW 264.7 cells.

### 2.3. Macrophage Cell Morphology in Different Ti-Discs Cultures

Cell shape has been lately considered an emerging property of the subtle interplay between cellular phenotype and physical properties [37]. As shown in Figure 4, we performed fluorescence microscopy to explore the morphology–phenotype connection in RAW 264.7 macrophages cultured on different titanium surfaces. RAW 264.7 macrophage cells seeded on the Ti-discs subjected to MA exhibited many cytoplasmic extensions and filopodia distinctive of macrophage activation [38,39]. However, macrophages grown on the other discs had a rounded morphology consistent with non-activated cells. Interesting, while the number of cells attached to the four different Ti surfaces was not significantly different, macrophages cultures on SB appeared to be more difficult to observe, more probably due to the high roughness of the surface and some impairment in cell adhesion. Polarity and wettability among different Ti surfaces had been reported [36,40] and these might affect the adherence performance of the macrophages. 

### 2.4. Determination of Cytokine Levels

Morphological features are not formal indications of macrophage biological responses. Thus, in order to evaluate the link between a potential priming after macrophage exposure to different Ti surface discs and the subsequent response triggered by TLR ligands stimulation, we determined IL-6 and IL-10 release by non-stimulated and stimulated macrophages with LPS (TLR4), imiquimod (TLR7), poly(I:C) (TLR3) and Pam3Csk4 (TLR2/1). 

Interestingly, none of the Ti surfaces studied and irrespective of the cell morphology appearance induced significant levels of IL-6 secretion in resting conditions (Figure 5A). Therefore, none of the Ti surfaces were able to shift macrophages towards a well-established pro-inflammatory status. However, Figure 5A showed a significant increase in IL-6 secretion (4396 pg·mL^−1^; *p* < 0.0001) by macrophages stimulated with LPS cultured on MA discs compared to SB + AE, SB and AE. Moreover, MA-cultured macrophages stimulated with imiquimod produced a two-fold IL-6 release (286 pg·mL^−1^; *p* < 0.0001) compared with the other Ti-surface discs. In a similar way, the Pam3Csk4 stimulation of MA-cultured macrophages also doubled IL-6 secretion compared with the other Ti-surfaces analyzed. Additionally, Poly(I:C) stimulation produced a three-fold induction of IL-6 secretion in MA-cultured macrophages (125.29 pg·mL^−1^) compared with the other Ti-based discs (*p* = 0.0057, *p* = 0.0024 and *p* = 0.0015, respectively).

Similarly, the resting levels of IL-10 secretion were not modified in macrophage cultures in MA, SB + AE, SB or AE (Figure 5B). After LPS-challenge, IL-10 release only revealed a significant increase of 2944 pg·mL^−1^ in MA-cultured macrophages compared to SB (*p* < 0.0001). Stimulation with imiquimod produced a two-fold release of IL-10 by MA-cultured macrophages (3340.42 pg·mL^−1^) compared to stimulated macrophages in the SB and AE discs (1375.6 pg·mL^−1^, *p* < 0.0001) and a likely effect was observed in IL-10 secretion stimulated with Pam3Csk4, where MA-cultured macrophages (1680 pg·mL^−1^) resulted in a two-fold induction secretion compared to SB discs (790.55pg·mL^−1^, *p* < 0.0001).

Thus, although Ti-based implants have a clear impact on the morphology of RAW 264.7 macrophages, this does not have a direct correlation as it might be expected in terms of macrophage skewing under resting conditions. However, the exposure to diverse Ti surfaces reprograms the macrophage to react very differently towards the same innate immune challenge. Although the overall observed effects remain the same, the quantification of additional cytokines could help to better understand the kinetics and/or TLR downstream signaling events modulated by Ti-discs in macrophages. Even though the molecular mechanisms remain to be elucidated, our results emphasize the importance of analyzing the dental implant material–macrophage interaction in situations in which the immune system must respond to unfold potential inflammatory or immunosuppressive effects that eventually might lead to chronic inflammation or autoimmunity processes.

### 2.5. Gene Expression of iNos/ArgI and Tlr Receptors in Macrophages Exposed to Different Ti-Discs

We next analyzed the expression pattern of *iNOS* and *ArgI* genes as M1 and M2 markers of macrophage polarization, respectively [41,42,43,44]. Macrophage exposure to different Ti-discs showed no induction of *iNos* and *ArgI* genes under resting conditions (Figure 6A,B). This is consistent with the IL-6 and IL-10 secretion pattern mentioned above. After TLR engagement with specific ligands, the highest levels of *iNOS* induction were achieved by macrophages exposed to different titanium-based implants and stimulated trough TLR4 with LPS and TLR3 with poly(I:C). Taken together, these results demonstrate that both MyD88-dependent (TLR4) and independent (TLR3) pathways [30] were not functionally affected by any of the different titanium-based materials studied in macrophages. As it was described above, the exposure of macrophages to distinct titanium-based materials differentially reprogrammed the cells to TLR-mediated responses. While MA-cultured macrophages displayed the highest level of *iNos* gene expression after TLR4 or TLR3 stimulation, the remaining implants (SB + AE, SB and AE) showed a statistically significant repression of *iNos* mRNA levels in the same experimental situation compared to MA surface. Although a similar pattern was observed as a result of TLR2/1 and TLR7 stimulation, the induction of *iNos* gene expression was lower than expected [45,46]. In addition, further studies should address the possibility of an impairment in the TLR2/1-mediated activation of the PI3K/Akt signaling after titanium-based implants that might partially explain our findings [47,48]. We next investigated the mRNA levels of *ArgI*. Since arginase and iNOS share a common substrate, L-arginine, arginase induction may regulate NO production by activated macrophages and thereby modulate the immune response [41,47]. However, the expression levels of *ArgI* mRNA were low and only detectable as a result of LPS stimulation (Figure 6B), being the highest level of expression after the MA exposure of macrophages. We cannot rule out an impairment in the MAPK pathway produced by the Ti-discs studied, as it has been suggested for other titanium-based materials [49], which might elucidate our findings as MAPK-dependent signaling is crucial for arginase induction in macrophages [50].

Finally, in order to gather further molecular insights, we next investigated whether the exposure of macrophages to the distinct Ti-discs regulated the mRNA levels of the genes for different TLRs. Our results showed a robust behavior in TLR steady-state mRNA levels, without statistically significant changes between the different experimental conditions for both control and TLR-stimulated cell cultures (Figure 7). Only the expression of TLR3 was modulated in poly(I:C)-stimulated macrophages, leading to a reduction in *Tlr3* mRNA levels in macrophage cell cultures for SB + AE (0.22, *p* < 0.0001), SB (0.33, *p* = 0.0397) and AE (0.13, *p* < 0.0001) compared to MA cell cultures. To the best of our knowledge, this is the first observation related to *Tlr* gene expression modulation in the context of different titanium-based dental implants. In this sense, it is worth mentioning that the endosomal TLR3 receptor is crucial for immune responses upon viral invasion and IRF3-dependent stimulation and chemotaxis of T cells [51,52]. Figure 8 summarizes the main findings of our work within the context of TLR signaling, suggesting potential mechanistic insights into the Ti-disc reprogramming of macrophages. 

## 3. Materials and Methods

### 3.1. Titanium-Based Materials

Titanium materials were commercial titanium alloy (Ti-6Al-4V ELI extra low interstitial medical grade) discs of 4 mm thickness and 12 mm diameter subjected to four different types of surface modifications: (MA) mechanical abrasion, (SB + AE) sandblasting plus acid etching, (SB) sandblasting and (AE) acid etching. Galimplant^®^ (Sarria, Spain) provided these different types of Ti materials subjected to surface modifications. SB and SB + AE were sandblasted with Al_2_O_3_ abrasive particles ranging from 250 to 450 μm at a pressure of 2.5 MPa. The distance from the gun to the surface was 100 mm. The acid etching was realized by an acid mixture 1:1 HCl and HNO_3_.

### 3.2. Chemical Composition of Ti-Discs

For the determination of the chemical composition of the different discs, a microanalysis in a XR energy dispersion spectrometer (Energy Dispersive Spectrometer EDS, Oxford, UK) was carried out.

### 3.3. Scanning Electron Microscopy

Electron Microscopy images were acquired in a Field Emission Scanning Electron Microscope (Thermo-Fisher Scientific, FEI Teneo, Hillsboro, OR, USA). Two secondary electron detectors were used to evaluate the surface topography: the on-camera ETD and the T1 in lens.

### 3.4. Surface Imaging, Analysis and Metrology by Interferometer

A confocal scanning in a 3D optical profiler (Sensofar S-Neox, Barcelona, Spain) was carried out to evaluate the surfaces of the discs and to measure the surface height. Images were taken with an objective of 100× and analyzed with the software SensoMap (Digital Surf, Bensançon, France) using a cut-off of 8 µm. For the quantitative evaluation of the surface roughness of the four different samples, a non-contact optical method was used that allows measurements on 3D structures by using a wave superposition principle with a visible-wavelength light (white light). A magnification of 20× with a 1× FOV was used, obtaining an image size of 227 × 298 µm^2^. Nine areas were randomly selected on the middle part of the sample surface, and the average of each parameter evaluated was calculated by the software. The following image filtering was set to calculate Ra parameter. The profile parameters were recorded after applying a gaussian filter. 

### 3.5. Murine Macrophage Cell Culture

A RAW 264.7 cell line of murine macrophages was obtained from ATCC (American Type Culture Collection, Gaithersburg, MD, USA) and cultured in complete high glucose Dulbecco’s Modified Eagle’s Medium (DMEM) (Gibco, ThermoFisher Scientific, Waltham, MA, USA) supplemented with 10% of FBS (Fetal Bovine Serum, Gibco, ThermoFisher Scientific, Waltham, MA, USA) and 1% of penicillin/streptomycin (Sigma-Aldrich, St. Louis, MO, USA), at 37 °C in a 5% CO2 incubator. 

### 3.6. Cell Morphology by Fluorescence Microscopy

For the cell morphology analysis, 1 mL of RAW 264.7 macrophages at 50,000 cells·mL^−1^ were cultured over titanium discs into 24-well plates in complete high glucose Dulbecco’s Modified Eagle’s Medium (DMEM) (Gibco, ThermoFisher Scientific, Waltham, MA, USA) and incubated for 48 h at 37 °C. Subsequently, the discs with attached cells were washed with PBS, transferred to a new plate and incubated for another 24 h. After the incubation, the discs with cells were fixed in 4% paraformaldehyde for 1 h at room temperature (RT); washed thrice in PBS; incubated 20 min at RT with Phalloidin-Atto 488 (Sigma-Aldrich, St. Louis, MO, USA); diluted 1:100 times in PBS containing 3% BSA (Sigma-Aldrich, St. Louis, MO, USA) and 0.5% of the permeabilizing detergent Triton-X-100 (Sigma-Aldrich, St. Louis, MO, USA); washed thrice in PBS; incubated 5 min at RT with DAPI (Sigma-Aldrich, St. Louis, MO, USA); diluted 1:5000 times (1 μg·mL^−1^) in PBS, 3% BSA and 0.5% triton; and washed thrice in PBS. Finally, the discs were mounted and examined, fluorescence images were captured at 63× magnification from randomly chosen areas with a Leica AF600 inverted fluorescence microscope using the Leica LAS AF digital image processing software (Leica Microsystems, Wetzlar, Germany).

### 3.7. TLR Stimulation

TLR ligands were acquired from InvivoGen (San Diego, CA, USA) and prepared according to the producer’s instructions. The TLR ligands used in this study were: bacterial lipopolysaccharide (LPS), synthetic bacterial lipoprotein Pam3Csk4 (Pam3Csk4), Imiquimod (Imiq) and low molecular weight polyinosine-polycytidylic acid (Poly(I:C-LMW)) (Poly(I:C)). RAW 264.7 cells were seeded over titanium discs into 24-well plates in a final volume of 1 mL of complete DMEM at a concentration of 250.000 cells·mL^−1^. After a 48 h period at 37 °C in a 5% CO_2_ incubator, the discs were washed with PBS and transferred to a new 24-well plate. Then, attached cells were stimulated by replacing the medium with 1 mL of complete DMEM with each TLR ligand according to previous works with minor modifications [34,35]. The final concentrations of the ligands were: LPS, 1 μg·mL^−1^; Pam3, 1 μg·mL^−1^; Imiquimod, 5 μg·mL^−1^; and Poly (I:C) 50 μg·mL^−1^. Cell culture samples without TLR ligand stimulation were used as controls.

### 3.8. Cell Viability and Plasma Membrane Integrity by LDH Assay

Cells were sowed over titanium discs. They were incubated for 48 h and then, stimulated as previously described. After 24 h at 37 °C in a 5% CO_2_ incubator, the lactate dehydrogenase (LDH) cytotoxicity detection assay (Roche, Basel, Switzerland) was carried out according to the manufacturer’s instructions.

### 3.9. Cytokine Release

RAW 264.7 cells were seeded as already described over Ti-discs into 24-well plates and incubated for 48 h. After stimulation, cells were incubated for another 24 h. Then, supernatants were centrifuged at 1000× *g* for 5 min at 4 °C and cell-cleared supernatants were collected and stored at −20 °C. IL-6 and IL-10 (OptEIA Mouse IL-6 and IL-10 sets, BD Pharmingen, San Diego, CA, USA) production was determined using enzyme-linked immunosorbent assay (ELISA) following the producer’s instructions. Chemokine levels presented in this article were performed with sample duplicates and are representative of ≥ 3 independent experiments.

### 3.10. Relative Gene Expression of mRNAs

RAW 264.7 cells were treated as described before with LPS, Pam3Csk4, Imiquimod and Poly(I:C) and incubated for 24 h at 37 °C with 5% CO_2_. After the treatment, the supernatants were collected and total RNA was extracted by adding 0.5 mL of PRImeZOL Reagent RNA purification kit (Canvax Biotech, Cordoba, Spain) to each well, following manufacturer’s instructions, and quantified using Nanodrop (ThermoFisher Scientific, Waltham, MA, USA). Samples were collected and stored at −20°C. RNA (1 ug) was reverse-transcribed by using PrimeScrip RT reagent kit with gDNA Eraser (ThermoFisher Scientific, Waltham, MA, USA), following the manufacturer’s procedure. The expression levels of the genes for hypoxanthine-phophoribosyltransferase (HPRT), iNOS, ArgI and TLRs 1, 2, 3, 4 and 7 [44] were assessed using quantitative real-time PCR (qRT-PCR method) with iTaq Universal SYBR Green Supermix (Bio-Rad, Hercules, CA, USA). Custom DNA oligos were designed to anneal in different exons as follows: HPRT (forward, 5′-GTAATGATCAGTCAACGGGGGAC-3′ and reverse, 5′-CCAGCAAGCTTGCAACCTTAACCA-3′), ArgI (forward, 5′-AACACGGCAGTCGCTTTAACC-3′ and reverse, 5′-GGTTTTCATGTGGCGCATTC-3′) and iNOS (forward, 5′-GCCTCATGCCATTGAGTTCATCAACC-3′ and reverse, 5′-GAGCTGTGAATTCCAGAGCCTGAAG-3′) were synthesized from IDT (Integrated DNA Technologies, Coralville, IA, USA). TLR1, TLR2 and TLR3 primers were purchased from Qiagen (Hilden, Germany). Primers for TLR4 (forward, 5′-ACCAGGAAGCTTGAATCCCT-3′ and reverse, 5′-TCCAGCCACTGAAGTTCTGA-3′) and TLR7 (forward, 5′-TCAAAGGCTCTGCGAGT-3′ and reverse, 5′-AGTCAGAGATAGGCCAGGA-3′) were purchased from Sigma (Sigma-Aldrich, St. Louis, MO, USA).

### 3.11. Data Analysis

All values are expressed as the mean ± standard deviation (SD). Each sample size is indicated in each figure legend. Statistical significance was evaluated by one-way analysis of variance (ANOVA) and Turkey’s multiple comparisons test using the GraphPad Prism 8.0. software (GraphPad Software Inc., San Diego, CA, USA).

## 4. Conclusions

Beyond studies on macrophage polarization and biocompatibility after exposure to different titanium-based dental implants, there are no comprehensive reports about macrophage functional responses in the context of TLR-mediated stimulation. Our results disclosed, for the first time, a reprogramming effect in macrophages as a result of different titanium surfaces widely used in dental applications on macrophages. Although titanium-driven morphological changes were evident, they did not differentially impact macrophages in resting conditions among the surfaces studied. However, the macrophages were differentially imprinted and displayed surface-dependent behaviors in terms of soluble immune mediators, mRNA expression of iNos and ArgI and TLR gene expression upon the selective activation of TLR2/1, TLR4, TLR3 and TLR7 signaling. These observations emphasize the need to study the outcome of immune responses under physiological stimulations to fully understand surface/macrophage interactions.

## Figures and Tables

**Figure 1 ijms-23-04285-f001:**
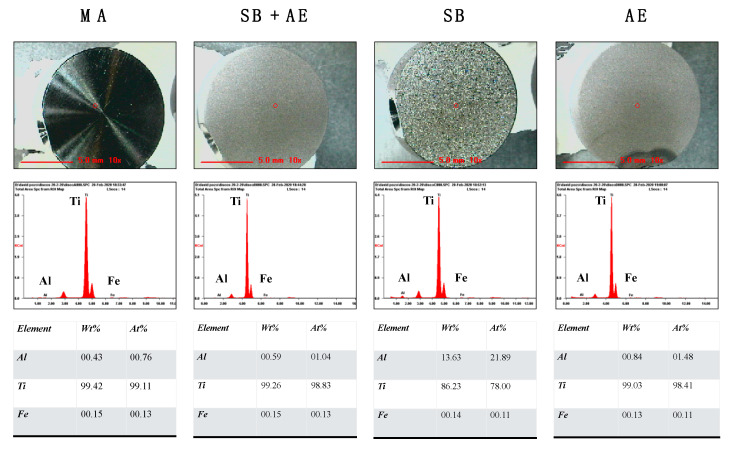
Composition of the different titanium discs by XR microfluorescence. The surface treatments were: mechanical abrasion (MA), sandblasting plus acid etching (SB + AE), sandblasting (SB) and acid etching (AE).

**Figure 2 ijms-23-04285-f002:**
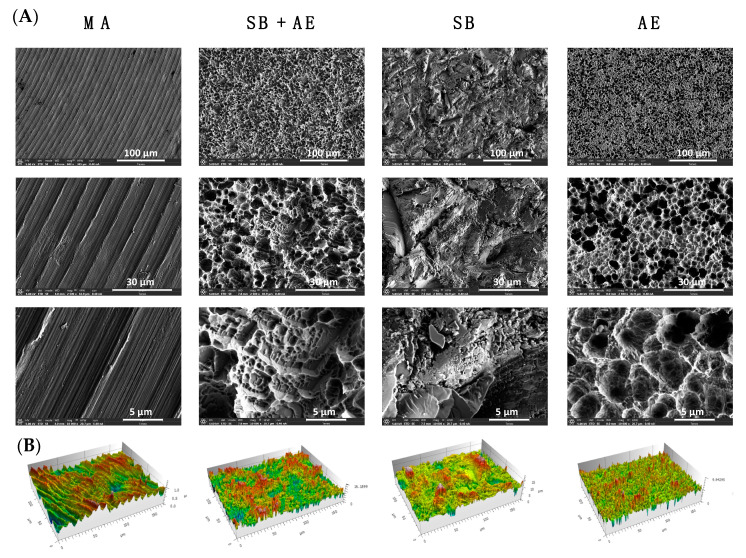
Ti-disc analysis (**A**) by FESEM and confocal scanning of the titanium discs with different surface treatment: mechanical abrasion (MA), sandblasting plus acid etching (SB + AE), sandblasting (SB) and acid etching (AE). Below, 3D representations of the surface analysis by optical profiling (**B**).

**Figure 3 ijms-23-04285-f003:**
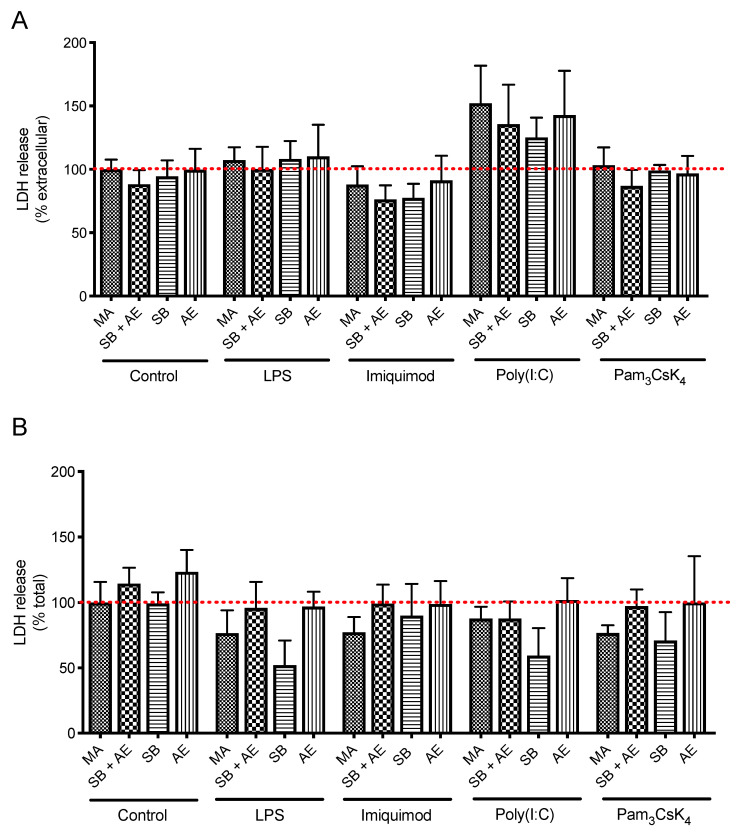
LDH extracellular (**A**) and total (**B**) quantification of RAW 264.7 cells cultured over the Ti-discs with different surfaces: mechanical abrasion (MA), sandblasting plus etching (SB + AE), sandblasting (SB) and acid etching (AE) and with or without the TLR ligands LPS (TLR4), imiquimod (TLR7), poly(I:C) (TLR3) and Pam3Csk4 (TLR2/1) (n = 6).

**Figure 4 ijms-23-04285-f004:**
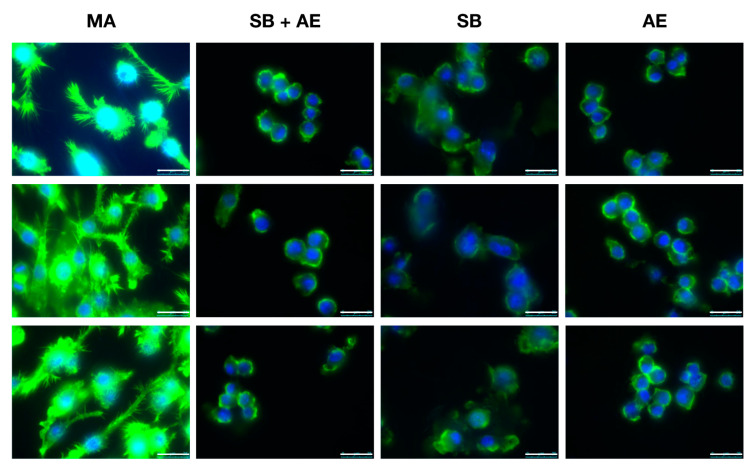
Cell morphology of RAW 264.7 cells cultured over the different Ti-discs by fluorescence microscopy. Surface treatment: mechanical abrasion (MA), sandblasting plus etching (SB + AE), sandblasting (SB) and acid etching (AE). In blue, nuclei with DAPI; in green, actin filaments with phalloidin 488. Scale bar: 25 µm. (n = 3).

**Figure 5 ijms-23-04285-f005:**
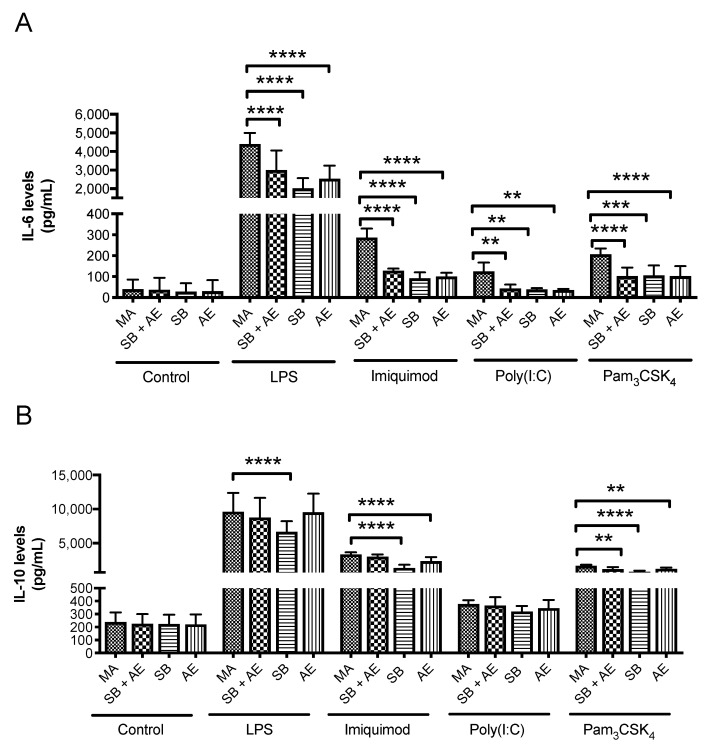
Quantitative determination by ELISA of IL-6 (**A**) and IL-10 (**B**) cytokine levels in supernatants from RAW 264.7 cells cultured over Ti-discs with different surfaces: mechanical abrasion (MA), sandblasting plus etching (SB + AE), sandblasting (SB) and acid etching (AE) and with or without the TLR ligands LPS (TLR4), imiquimod (TLR7), poly(I:C) (TLR3) and Pam3Csk4 (TLR2/1) (n = 3). **, *p* < 0.0021; ***, *p* < 0.0002; ****, *p* < 0.0001.

**Figure 6 ijms-23-04285-f006:**
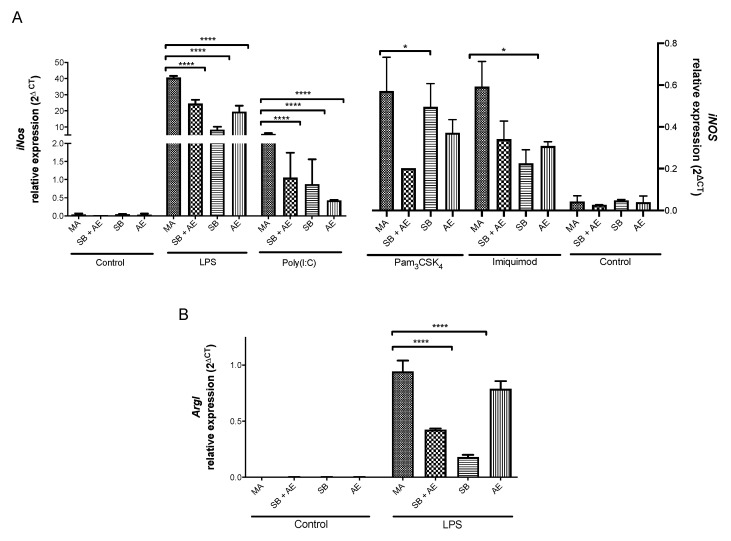
Relative gene expression of *iNos* (**A**) and *ArgI* (**B**) mRNAs by qRT-PCR in RAW 264.7 cells cultured over Ti-discs with different surfaces: mechanical abrasion (MA), sandblasting plus etching (SB + AE), sandblasting (SB) and acid etching (AE) and with or without the TLR ligands LPS (TLR4), Imiquimod (TLR7), Poly(I:C) (TLR3) and Pam3Csk4 (TLR2/1). * *p* < 0.00332; **** *p* < 0.0001.

**Figure 7 ijms-23-04285-f007:**
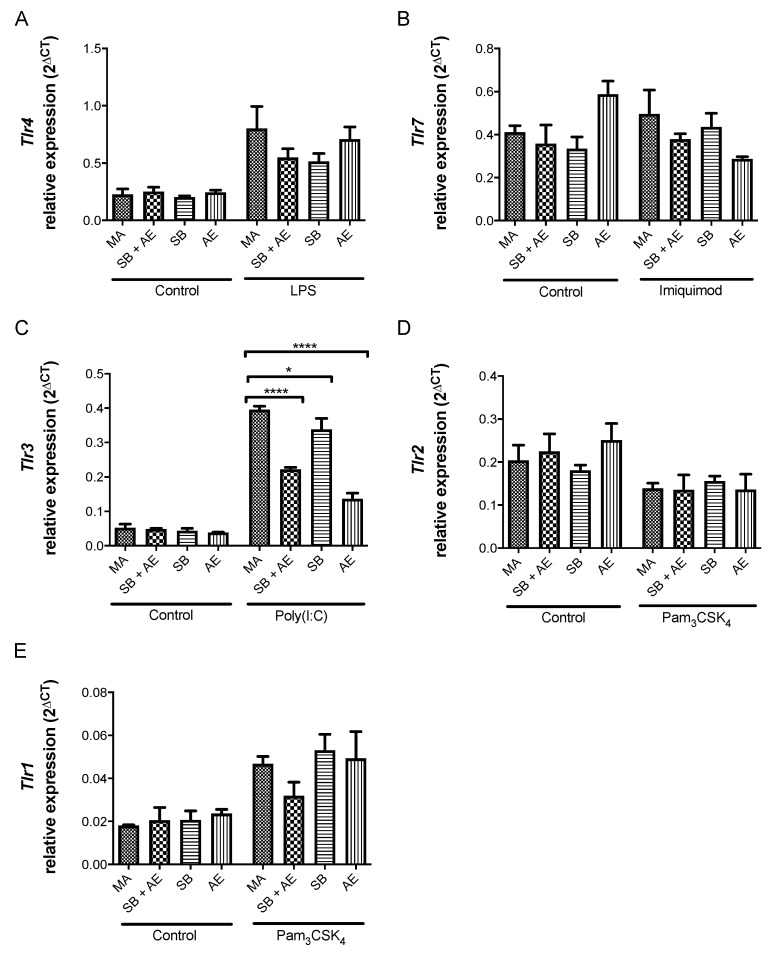
Relative gene expression of *Tlr 4* (**A**), *Tlr7* (**B**), *Tlr3* (**C**), *Tlr2* (**D**) and *Tlr1* (**E**) mRNAs by qRT-PCR in RAW 264.7 cells cultured over the Ti-discs with different surfaces: mechanical abrasion (MA), sandblasting plus etching (SB + AE), sandblasting (SB) and acid etching (AE) and with or without the TLR ligands LPS (TLR4), Imiquimod (TLR7), Poly(I:C) (TLR3) and Pam3Csk4 (TLR2/1). *, *p* < 0.00332; ****, *p* < 0.0001.

**Figure 8 ijms-23-04285-f008:**
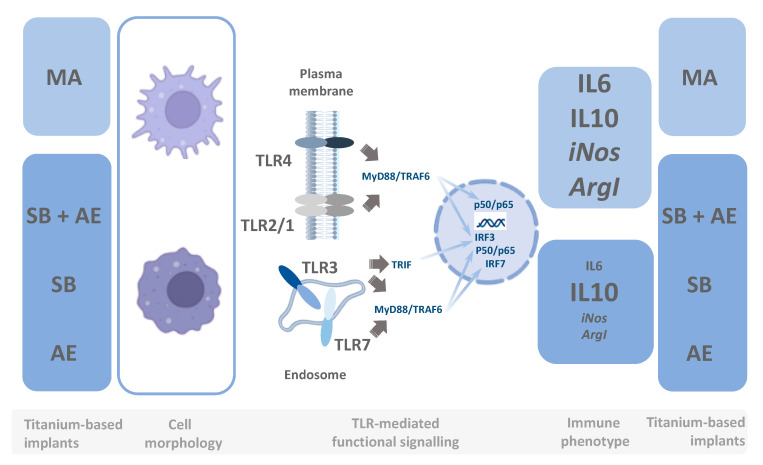
Potential mechanistic insights into Ti-discs caused the reprogramming of macrophages. TLR signaling imprinting of the MA-exposed macrophages displays differential functional responses compared to SB + AE, SB and AE titanium-based materials after membrane or endosomal TLRs elicited by specific ligands (LPS for TLR4, synthetic bacterial triacylated lipoprotein for TLR2/1, poly I:C for TLR3 and imiquimod for TLR7). TLR downstream signaling regulated by titanium-based dental materials affects both the MyD88-dependent and MyD88-independent pathways, suggesting a potential role in IRF3-, IRF7- and p50/p65-mediated control of the transcriptional activity. Abbreviations: MyD88, MyD88 innate immune signal adaptor protein; TRAF6, TNF receptor-associated factor 6; TRIF, TIR-domain-containing adapter-inducing interferon-β; IRF3, interferon regulatory factor 3; IRF7, interferon regulatory factor 7; p50/p65, subunits of NF-kappa B nuclear transcription factor; MA, mechanical abrasion; SB, sandblasting plus etching; SB, sandblasting; AE, acid etching.

**Table 1 ijms-23-04285-t001:** Mean ± standard deviation of the surface roughness parameter Ra for the different types of Ti implants.

Ti-Disc	MA	SB + AE	SB	AE
R_a_ (µm)	0.026 ± 0.012	1.235 ± 0.024	0.932 ± 0.523	0.603 ± 0.181

## Data Availability

The data presented in this study are available on request from the corresponding author.

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
