# Peer review of "Titanium Surface Characteristics Induce the Specific Reprogramming of Toll-like Receptor Signaling in Macrophages"

_ijms, 2022, doi:10.3390/ijms23084285_

Round 1
Reviewer 1 Report
Tha manuscript "Toll-like receptor signalling in macrophages is modulated by dental titanium discs: linking implants surface characteristics and innate immunity phenotypes" is focus on the characterization of the effect of functionalizing four different Ti surfaces (mechanical abrasion, sandblasting plus etching; sandblasting and acid etching) with four TRLs ligands on the biological response of RAW 264.7 macrophages in vitro.
The result of this work may be of interest to the readers of the journal and the manuscript is properly written. However, there are some points in the work that should be clarified and improved.
- In materials and methods section, more information about surface treatment preparation should be added.
- In materials and methods section, the sequence of the targets evaluated by qRT-PCR should be included.
- The authors could explain the reason why IL6 cytoquine was chosen as a marker associated with a pro-inflammatory polarization. Cytokines such as IL1b and TNFa play a fundamental role in this regards and are also normally used as markers study macrophage responses. Only a partial view of the RAW 264.7 response to TRLs can be obtained if just IL6 secretion is measured. This experimental limitation should be discussed.
Author Response
Dear Reviewer,
Thanks for taking the time to review our manuscript and suggest to us to improve our work by providing a lot more detail. We have done so, and we are now submitting a manuscript that not only addresses the points the you specifically raised but also many others that we have considered in order to deliver what we think is a much improved version of our work. This version includes more paragraphs, English grammar revisions in all main sections, new references. Thanks a lot. We are looking forward to your comments.
Sincerely,
Francisco-Javier Gil Mur
Reviewer 1
Tha manuscript "Toll-like receptor signalling in macrophages is modulated by dental titanium discs: linking implants surface characteristics and innate immunity phenotypes" is focus on the characterization of the effect of functionalizing four different Ti surfaces (mechanical abrasion, sandblasting plus etching; sandblasting and acid etching) with four TRLs ligands on the biological response of RAW 264.7 macrophages in vitro.
The result of this work may be of interest to the readers of the journal and the manuscript is properly written. However, there are some points in the work that should be clarified and improved.
- In materials and methods section, more information about surface treatment preparation should be added.
The authors have incoporated more information in the materials and methods and the results and discussion.
- In materials and methods section, the sequence of the targets evaluated by qRT-PCR should be included.
We do appreciate the reviewer’s comments. Despite references included, the sequence of the F/R primers were indicated as 5’-´3 for the corresponding annotated gene using standard terminology.
- The authors could explain the reason why IL6 cytoquine was chosen as a marker associated with a pro-inflammatory polarization. Cytokines such as IL1b and TNFa play a fundamental role in this regards and are also normally used as markers study macrophage responses. Only a partial view of the RAW 264.7 response to TRLs can be obtained if just IL6 secretion is measured. This experimental limitation should be discussed.
We do appreciate the reviewer’s suggestion/comment. There are several, not only TNFa or IL1b cytokines that play a major role in pro-inflammatory polarization of macrophages. Thus, we determined IL-6 as a pro-inflammatory cytokine due to the fact that is a responsive one for TLR ligands with a convenient dynamic range in our cell line setting. The reviewer is right in the sense that extra info on other inflammatory cytokines might provide a more detailed info. However, the overall effect will remain as described. Other cytokines such as the one suggested will likely have different kinetics, but in terms of secretion profile will be very similar. This is the reason why we measured IL-10 and the gene expression of other well-known markers. Of course, a comprehensive study including other inflammatory cytokines will give info related to potential molecular mechanisms affected by Ti-discs. This is valuable irrespective of the general readout and message. Therefore, we included a sentence to highlight this limitation and to encourage further exploration (Lines 248-253).
Reviewer 2 Report
General comment: The authors covered an interesting topic and organized the paper comprehensively! There are several points to ameliorate while reading. Also a new English check is needed to proceed. The respective comments are listed below:
- A suggestion for the title, to improve it, in order to be clearer and more engaging even for non-specialist colleagues.
- Enter the full name of the TRLs in line 26.
- In line 43, it is better to add after titanium the word “discs”, so “titanium discs”.
- In the introduction section, please improve the first sentence (lines 47-50). On line 72 put the acronym of TRLs into brackets after its full name; in line 84, it is necessary also to insert the respective acronyms into brackets of MA, SB+AE, etc…, being mentioned here for the first time. Line 86: put into brackets the acronym of (LPS) and of imiquimod, because the latter acronym is reported in the following paragraphs.
The referee suggests also to the authors considering these interesting studies: [Evaluation of Biological Response of STRO-1/c-Kit Enriched Human Dental Pulp Stem Cells to Titanium Surfaces Treated with Two Different Cleaning Systems. Int J Mol Sci. 2019 Apr 16;20(8):1868. doi: 10.3390/ijms20081868. PMID: 31014017]; [Differential Efficacy of Two Dental Implant Decontamination Techniques in Reducing Microbial Biofilm and Re-Growth onto Titanium Disks In Vitro. Appl. Sci. 2019, 9, 3191. https://doi.org/10.3390/app9153191]
- In line 95 put the full name of FESEM first and also keep the acronym of XR during the whole manuscript, because in several parts you find X-Ray and in other XR. In lines 96-97-98, delete the full the names and maintain only the acronyms of MA and etc… In line 99 delete “in” before titanium because doesn’t make sense or improves the entire sentence; also, here insert before the full name of the symbol of Al. In line 162 put the acronym of titanium “Ti-discs”; in line 174 put the full name of LDH before and then its acronym.
- Please change the format of the columns in figures 3,5,6,7 and making it same for each ligand, for example the column of MA must be the same as the format for the control, LPS and so on; in the same way also for the other columns. At present, these seem to be different each time…
- In line 251 we observe the expression "titanium-based dental implants" which in reality is replaced in "Ti-discs" ... the referee suggests inserting the acronym "Ti-discs" in brackets after the complete expression and keeping this latter throughout the manuscript, because it creates confusion that at the beginning it is another expression and, at the end, another… This can also be corrected in line 20 (abstract).
- Line 255: what is the meaning of “trough” here in this sentence…? Maybe you wanted to write “through”…?
- Lines 292-295: Please improve this sentence!
- Line 302: A suggestion to correct “mechanistic” into “mechanical”.
- In line 340, what does miss here in the place of “_”?
- In line 346, put before the full name of DMEM and then its acronym into brackets.
- In subsection 3.6, it is necessary to indicate in brackets for each device/apparatus the respective manufacturer, city and state of the same; the same in line 396.
- In line 366, the CO2 needs to be corrected in “CO2”.
Author Response
Dear Reviewer,
Thanks for taking the time to review our manuscript and suggest to us to improve our work by providing a lot more detail. We have done so, and we are now submitting a manuscript that not only addresses the points the you specifically raised but also many others that we have considered in order to deliver what we think is a much improved version of our work. This version includes more paragraphs, English grammar revisions in all main sections, new references. Thanks a lot. We are looking forward to your comments.
Sincerely,
Francisco-Javier Gil Mur
Reviewer 2
The authors covered an interesting topic and organized the paper comprehensively! There are several points to ameliorate while reading. Also a new English check is needed to proceed.
We do thanks the positive feedback from the reviewers. As suggested, a careful English check has been carried out.
All the changes are highlighted in blue for inspection.
The respective comments are listed below:
- A suggestion for the title, to improve it, in order to be clearer and more engaging even for non-specialist colleagues.
We do appreciate the reviewer’s comments. Although no alternative title was suggested, we modified the title towards a broader, more inclusive one with a more general meaning in relation to the material-cell interaction in the context of the innate response mediated by TLRs in macrophages. The new title is as follows “Titanium surface characteristics induce specific reprogramming of the Toll-like receptor signaling in macrophages”.
- Enter the full name of the TRLs in line 26.
We appreciate the reviewer’s indication and it has been edited accordingly in the revised version.
- In line 43, it is better to add after titanium the word “discs”, so “titanium discs”.
We appreciate the reviewer’s indication and it has been edited accordingly in the revised version.
- In the introduction section, please improve the first sentence (lines 47-50).
We appreciate the reviewer’s suggestion. The para has been edited to improve the understanding of the message as it follows:
“The last few years have witnessed a growing demand for implantable medical devices, mainly for two reasons. On the one hand, important advances in technological and industrial development in nano- and micro-structured materials and on the other, an aging world population with more chronic diseases and greater needs for these alternatives [1,2].”
On line 72 put the acronym of TRLs into brackets after its full name;
Thanks for noticing us. It has been edited accordingly.
in line 84, it is necessary also to insert the respective acronyms into brackets of MA, SB+AE, etc…, being mentioned here for the first time.
Thanks for noticing us. It has been edited accordingly.
Line 86: put into brackets the acronym of (LPS) and of imiquimod, because the latter acronym is reported in the following paragraphs.
We appreciate the reviewer’s indication and it has been edited accordingly in the revised version.
The referee suggests also to the authors considering these interesting studies:
We do appreciate the reviewer’s suggestion. After reading the content of both papers is kind of difficult to include them in our discussion or introduction sections. They are dealing with decontamination procedures to remove bacterial biofilm in titanium dental implants in the context of surface alteration and/or biological features of dental pulp stem cells. The common niche in our context might be the material but is difficult to link to innate immune responses as decontamination procedures work as physical/chemical disruptors of biofilm. Anyhow, it has been nice to have them as we are included them in another manuscript to be submitted soon dealing with nanoparticle-DOX and the relation to macrophage polarization. In this context of antibiotic activity, both papers fit well.
[Evaluation of Biological Response of STRO-1/c-Kit Enriched Human Dental Pulp Stem Cells to Titanium Surfaces Treated with Two Different Cleaning Systems. Int J Mol Sci. 2019 Apr 16;20(8):1868. doi: 10.3390/ijms20081868. PMID: 31014017]; [Differential Efficacy of Two Dental Implant Decontamination Techniques in Reducing Microbial Biofilm and Re-Growth onto Titanium Disks In Vitro. Appl. Sci. 2019, 9, 3191. https://doi.org/10.3390/app9153191]
- In line 95 put the full name of FESEM first and also keep the acronym of XR during the whole manuscript, because in several parts you find X-Ray and in other XR.
We appreciate the reviewer’s indication and it has been edited accordingly in the revised version.
In lines 96-97-98, delete the full the names and maintain only the acronyms of MA and etc…
We appreciate the reviewer’s indication and it has been edited accordingly in the revised version. We left the full details only at figure legends to facilitate an independent reading of graphic material versus text material of the manuscript.
In line 99 delete “in” before titanium because doesn’t make sense or improves the entire sentence; also, here insert before the full name of the symbol of Al.
We appreciate the reviewer’s indication and it has been edited accordingly in the revised version.
In line 162 put the acronym of titanium “Ti-discs”; in line 174 put the full name of LDH before and then its acronym.
We appreciate the reviewer’s indication and it has been edited accordingly in the revised version.
- Please change the format of the columns in figures 3,5,6,7 and making it same for each ligand, for example the column of MA must be the same as the format for the control, LPS and so on; in the same way also for the other columns. At present, these seem to be different each time…
Thanks for noticing us. All figures (3,5,6 and 7) have been edited accordingly. Figures have gained in clarity. We do appreciate it.
- In line 251 we observe the expression "titanium-based dental implants" which in reality is replaced in "Ti-discs" ... the referee suggests inserting the acronym "Ti-discs" in brackets after the complete expression and keeping this latter throughout the manuscript, because it creates confusion that at the beginning it is another expression and, at the end, another… This can also be corrected in line 20 (abstract).
We appreciate the reviewer’s indication. The text referred in line 251 is a general discussion not restricted to Ti-discs. In this sense, our work do emphasize the importance of analyzing the material-cell interaction in terms of functional innate immune responses and not merely in resting cells. A fact usually overlooked in the literature. This is why we stated “dental implant material-macrophage” (not “titanium-based dental implants”) on purpose. We do agree to homogenize the terminology for the titanium-based dental implants throughout the text in the remaining cases when is exclusively referred to the Ti-discs analyzed and, therefore, we changed accordingly in line 20 (abstract) and elsewhere.
- Line 255: what is the meaning of “trough” here in this sentence…? Maybe you wanted to write “through”…?
Thanks for the comment. The meaning is correct. The sentence starts as “Even though…” being the term a synonymous of “Although…”, “Despite the fact…”
- Lines 292-295: Please improve this sentence!
We appreciate the reviewer’s indication and it has been edited accordingly in the revised version.
- Line 302: A suggestion to correct “mechanistic” into “mechanical”.
We do appreciate the reviewer’s comment. The term “mechanical” refers to aspects related to physical phenomenology while the term “mechanistic” refers to molecular and/or cellular mechanisms of action. In this sense, it is widely used in molecular biology/biochemistry and in that way was applied in the text.
- In line 340, what does miss here in the place of “_”?
We appreciate the reviewer’s indication. It was a typo and it has been edited accordingly. Now, cell line provider (ATCC) is indicated.
- In line 346, put before the full name of DMEM and then its acronym into brackets.
We appreciate the reviewer’s indication and it has been edited accordingly in the revised version.
- In subsection 3.6, it is necessary to indicate in brackets for each device/apparatus the respective manufacturer, city and state of the same; the same in line 396.
We appreciate the reviewer’s indication and it has been edited accordingly in the revised version. Also in section 3.5 and elsewhere in the whole MM section.
- In line 366, the CO2 needs to be corrected in “CO2”.
We appreciate the reviewer’s indication and it has been edited accordingly in the revised version. It has been edited in Line 389.